# The Role of Mycotoxins in Interactions between *Fusarium graminearum* and *F. verticillioides* Growing in Saprophytic Cultures and Co-Infecting Maize Plants

**DOI:** 10.3390/toxins15090575

**Published:** 2023-09-18

**Authors:** Mohammed Sherif, Nadine Kirsch, Richard Splivallo, Katharina Pfohl, Petr Karlovsky

**Affiliations:** 1Molecular Phytopathology and Mycotoxin Research, University of Göttingen, 37077 Göttingen, Germany; 2Institute for Molecular Biosciences, Goethe University Frankfurt, 60438 Frankfurt am Main, Germany; 3Phytopathology Unit, Plant Protection Department, Desert Research Center, Cairo 11753, Egypt; 4Institute for National and International Plant Health, Julius Kühn-Institut, 38104 Braunschweig, Germany; 5Nectariss Grasse SAS, 06130 Grasse, France

**Keywords:** *Fusarium*, mixed infection, fungal competition, deoxynivalenol, nivalenol, fumonisins

## Abstract

*Fusarium graminearum* (*FG*) and *Fusarium verticillioides* (*FV*) co-occur in infected plants and plant residues. In maize ears, the growth of *FV* is stimulated while *FG* is suppressed. To elucidate the role of mycotoxins in these effects, we used *FG* mutants with disrupted synthesis of nivalenol (NIV) and deoxynivalenol (DON) and a *FV* mutant with disrupted synthesis of fumonisins to monitor fungal growth in mixed cultures in vitro and in co-infected plants by real-time PCR. In autoclaved grains as well as in maize ears, the growth of *FV* was stimulated by *FG* regardless of the production of DON or NIV by the latter, whereas the growth of *FG* was suppressed. In autoclaved grains, fumonisin-producing *FV* suppressed FG more strongly than a fumonisin-nonproducing strain, indicating that fumonisins act as interference competition agents. In co-infected maize ears, *FG* suppression was independent of fumonisin production by *FV*, likely due to heterogeneous infection and a lower level of fumonisins in planta. We conclude that (i) fumonisins are agents of interference competition of *FV*, and (ii) trichothecenes play no role in the interaction between *FG* and *FV*. We hypothesize the following: (i) In vitro, *FG* stimulates the *FV* growth by secreting hydrolases that mobilize nutrients. In planta, suppression of plant defense by *FG* may additionally play a role. (ii) The biological function of fumonisin production in planta is to protect kernels shed on the ground by accumulating protective metabolites before competitors become established. Therefore, to decipher the biological function of mycotoxins, the entire life history of mycotoxin producers must be considered.

## 1. Introduction

### 1.1. Maize pathogens Fusarium graminearum and F. verticillioides

Maize (*Zea mays* L.) is a major commodity crop, with more than 1 billion tons harvested per year [1]. As compared to their wild ancestors, maize is more susceptible to fungal pathogens [2], which reduce the yield and contaminate the grain with mycotoxins. Fungi of the genus *Fusarium* are among the major pathogens infecting maize.

The most important *Fusarium* species infecting maize ears are *F. graminearum*, which dominates in moderate and cold climate, and *F. verticillioides*, which dominates in warm areas. The infection of maize ears with *F. graminearum*, the causal agent of *Gibberella* ear rot, can be recognized through pinkish or reddish fungal mycelium on kernels. It typically begins at the tip of the cob [3] and requires temperatures between 24 and 26 °C and high humidity [4]. Infection of maize with *F. verticillioides*, the causal agent of *Fusarium* ear rot, results in sporadic colonization of kernels with the appearance of whitish mycelium in the late stages. It is favored by high temperatures and dry conditions [4,5,6]. Asymptomatic infection of maize ears with *F. verticillioides* and endophytic growth of the fungus have often been reported [7,8,9]. Endophytic growth of *F. graminearum* has only been reported in plants different from maize [10,11]. Maize ears are often co-infected with several *Fusarium* species [12], yet interactions among *Fusarium* pathogens in maize tissue are poorly understood.

*F. graminearum* and *F. verticillioides* infect maize ears by distinct routes. Both pathogens infect at flowering when maize ears develop silks. *F. graminearum* grows in and on the silks, penetrates the ovaries, colonizes the space between kernels, and eventually reaches the rachis [13]. Since Koehler [14] in the 1940s, silks have been known to be the most important entry for the infection of maize with *F. verticillioides* [4,5,9,15]. While many researchers have carried out infection assays by inoculating silks, others have been unable to achieve kernel infection by spraying silks with conidia [16]. Another popular inoculation method has been to inject conidia between the husks with a hypodermic needle. Inoculating a *F. verticillioides* strain expressing a fluorescent protein by this method, Duncan and Howard [16] observed that the fungus entered maize kernels through the styler canal, which is a natural opening on the kernel’s surface located below the site of silk attachment. *F. verticillioides* can also infect maize asymptomatically [7,9,17,18]. Scanning electron microscopy of matured asymptomatic maize kernels infected with *F. verticillioides* revealed the presence of fungal mycelium in intercellular space of kernel pedicles [19].

### 1.2. Trichothecenes of Fusarium graminearum and Fumonisins of F. verticillioides and Their Role in Plant Disease

Both *F. graminearum* and *F. verticillioides* produce mycotoxins. Mycotoxins of *F. graminearum* include type B trichothecenes (Figure A1) nivalenol (NIV), deoxynivalenol (DON), and their acetylated derivatives. *F. graminearum* strains are assigned to chemotypes according to the major trichothecene they produce [20]. Contrary to previous assumptions, all strains are thought to produce both DON and NIV [21], with chemotypes corresponding to trichothecenes produced in the largest amounts. Trichothecenes are a health concern [22], and despite considerable research efforts, they pose a threat to food safety [20]. Trichothecenes of *F. graminearum* are virulence factors in wheat spikes and maize ears but not in barley spikes [23,24]. NIV, but not DON, acts as a virulence factor in maize ears [25].

*F. verticillioides* produces mainly fumonisins, fusarin C, and fusaric acid. Fumonisins (Figure A1) are important due to their frequent occurrence and multiple toxicity, including carcinogenicity [26]. Fumonisins are also phytotoxic [27], with the amine group being critical for toxicity [28]. The role of fumonisins in disease development has been controversial. Fumonisins were detected in asymptomatic maize kernels (reviewed by Munkvold [4]), indicating that they do not induce disease symptoms. Stimulation of fumonisin synthesis in maize kernels suggests that fumonisins play a role in the colonization of plant tissue [29], but experiments with mutants of *F. verticillioides* with disrupted fumonisin synthesis showed that fumonisins were not required for the infection of maize ears [30,31,32]. Inoculation of two mutants of *F. verticillioides* not producing fumonisins and their wildtype progenitors into the roots of maize, sorghum, and other plants also did not reveal any effect of fumonisins on plant colonization [18]. Contradictory findings were reported by Williams et al. [33] and Glenn at al. [34]. Williams et al. compared wildtype strains differing in the production of fumonisins, but we do not consider these results conclusive, since the strains likely differed in other characteristics as well, which could account for the observed differences in aggressiveness. Glenn at al. [34] provided convincing evidence for the role of fumonisins in seedling disease by restoring fumonisin synthesis in a banana strain lacking fumonisin synthesis. One possible reason for the conflicting results on maize seedlings, apart from using different maize varieties, could be that Glenn et al. inoculated the seeds before sowing, allowing the pathogen to establish before plant defense responses could unfold [34], while Dastjerdi and Karlovsky [18] inoculated three-week-old seedlings by dipping their roots in spore suspension.

### 1.3. Interaction between F. graminearum and F. verticillioides during Saprophytic Growth and the Role of Mycotoxins Produced by These Fungi

Saprophytic growth of plant-pathogenic *Fusarium* spp. constitutes a major part of their life cycle and is instrumental in inoculum formation, yet our understanding of the saprophytic phase of necrotrophic pathogens is limited, as pointedly noted by Audenaert et al. [35], because research on fungal pathogens in phytopathology has focused mainly on fungus–plant interactions. The first study addressing the competition between *F. graminearum* and *F. verticillioides* in vitro was published by Marín et al. in 1998 [36]. While monitoring fungal colonies growing on agar, the authors observed that the growth of *F. verticillioides* was substantially reduced by *F. graminearum* regardless of water activity while the growth of *F. graminearum* was not affected by the presence of *F. verticillioides.* This was exactly the opposite of what was observed in planta (see the next section). Velluti et al. obtained similar results [37,38] by growing the fungi on thin layers of autoclaved maize grains inoculated with agar disks and counting fungal spores as a proxy for the biomass. The growth of *F. verticillioides* was suppressed by co-cultivation with *F. graminearum*, but the growth of *F. graminearum* was only slightly inhibited [37] or even stimulated [38] by the presence of *F. verticillioides*. A low competitive fitness of *F. verticillioides* in vitro was also reported for liquid co-cultures with *Aspergillus niger* [39] and solid co-cultures with *Aspergillus flavus* [40].

Defense against antagonists and predators is considered the most important biological function of mycotoxins [41,42]. Toxicity of mycotoxins of *Fusarium* spp. towards fungi is established for fumonisins [43,44] as well as trichothecenes [45,46,47]. The effect of co-cultivation with other fungi on mycotoxin synthesis [37,39,48] and the detoxification of *Fusarium* mycotoxins by filamentous fungi [49,50,51,52] further support the role of mycotoxins in fungus–fungus competition. Rigorous proofs exist for the role of *Fusarium* mycotoxins zearalenone in defense against mycoparasites [53] and aurofusarin in defense against predators [54]. On this background, it is reasonable to hypothesize that DON and fumonisins suppress the growth of competing fungi in mixed cultures.

### 1.4. Mixed Infection of Maize Ears with F. graminearum and F. verticillioides

When several pathogens infect the same plant, their secreted secondary metabolites may modulate the outcome of the infection not only by direct inhibition but also by the suppression or induction of the defense responses in the plant. A study of mixed infection of maize ears in 1999 in Canada [6] showed for the first time that co-infection of maize with *F. verticillioides* and *F. graminearum* suppressed the growth of the latter. The authors also indicated that the growth of *F. verticillioides* was facilitated by the co-infection with *F. graminearum*, but the results were obscured by spontaneous infections. A key study addressing mixed infection of maize with *F. graminearum* and *F. verticillioides* was carried out a few years later in France [55]. Despite the high variability inherent in field trials, the authors convincingly confirmed suppression of *F. graminearum* and stimulation of *F. verticillioides* growth in mixed infection. Both studies used silk channel inoculation, which keeps the kernels intact, preserving a mechanical barrier to infection. Fungal biomass in planta was quantified by species-specific real-time PCR. Co-inoculation of maize ears with *F. graminearum* and *F. verticillioides* was also investigated by Giorni et al. [56]. In their study, mechanical barriers to infection were disrupted by stabbing the ears with a fork; therefore, a comparison with previous work is difficult. Furthermore, according to the authors’ note “...among considered mycotoxins, only DON resulted to be influenced by the year since it resulted completely absent in year 2017”, the infection with *F. graminearum* was inefficient. The last study of maize ears co-inoculated with *F. graminearum* and *F. verticillioides* focused on the analysis of volatile metabolites [57]. The authors used the same inoculation method as the Canadian [6] and French [55] groups and largely corroborated their results, especially regarding the suppression of in planta growth of *F. graminearum* by co-inoculation with *F. verticillioides.* A high competitive fitness of *F. verticillioides* in planta was also reported for co-inoculation with *Aspergillus flavus*, as *F. verticillioides* suppressed the spread of *A. flavus* to the surrounding tissue [58]. In another study, *F. verticillioides* reduced the lesions caused by *A. flavus*, while the lesions produced by *F. verticillioides* were unaffected by the presence of *A. flavus* [59]. Endophytically growing *F. verticillioides* also suppressed the infection of maize plants with *Ustilago maydis* [60]. On the other hand, the competitive fitness of *F. graminearum* in planta was lower than that of co-inoculated endophytes *Epicoccum nigrum* or *Sordardia fimicola* [61], similar to the suppression of *Fusarium* head blight by traditional biocontrol agents [62]. An important observation in the study by Abdallah et al. [61] was that suppression efficiency varied greatly among strains of the same endophyte. 

### 1.5. Objectives

The objective of this work was to elucidate the role of mycotoxins of *F. graminearum* and *F. verticillioides* in interactions between these fungi in mixed saprophytic cultures and in mixed infection of maize ears, using mutants of *F. graminearum* with disrupted biosynthesis of trichothecenes and a mutant of *F. verticillioides* with disrupted biosynthesis of fumonisins.

## 2. Results

### 2.1. Competition between F. graminearum and F. verticillioides in Saprophytic Growth

As a common model of competition among fungi during saprophytic growth, pairs of mycotoxin-producing strains of *F. verticillioides* and *F. graminearum* and their mutants with disrupted toxin synthesis were inoculated on PDA plates and grown at 25 °C for 10 days (Figure 1). Pink-colored colonies of *F. graminearum* grew faster than pale colonies of *F. verticillioides.* No effect of the production of DON, NIV, or fumonisin on the interaction between the colonies was observed. Similarly, dual cultures with the same strain combinations on MMA plates (maize extract malt agar) have not revealed any effect of the synthesis of fumonisins, DON, or NIV.

Fungal growth on the surface of agar does not simulate well the conditions in plant debris. Therefore, we also examined how the two *Fusarium* species competed in autoclaved maize kernels inoculated with a spore suspension, which ensured that all kernels were infected with both species and facilitated intense fungus–fungus interactions. Fungal biomass was determined by species-specific qPCR. A disadvantage of this method is that, due to the exponential nature of PCR, small errors in the threshold cycle are magnified into large errors in the calculated DNA concentration. To reduce the error in estimated DNA concentration, we used a DNA extraction method designed to suppress sampling error, and calibration curves for qPCR were constructed with standards spread more densely than usual (see Section 5 and Appendix A Figure A2). The results (Figure 2) showed that the growth of *F. graminearum* was strongly suppressed in the presence of *F. verticillioides.* Furthermore, a *F. verticillioides* strain producing fumonisins suppressed the growth of *F. graminearum* more strongly than an isogenic strain unable to synthesize fumonisins. 

The growth of *F. verticillioides* in dual cultures with *F. graminearum* compared to single-strain cultures was stimulated. This effect was not affected by the production of trichothecenes by the co-inoculating *F. graminearum* strain (Figure 2). 

### 2.2. Disease Severity, Fungal Biomass, and Mycotoxin Accumulation in Maize Ears Inoculated with Single Strains of F. graminearum or F. verticillioides

Maize ears were inoculated separately with mycotoxin-producing strains of *F. verticillioides* and *F. graminearum* and their mycotoxin-nonproducing mutants 5 days after silking. Disease severity was evaluated 18 days after inoculation. Control treatments (mock inoculation with water) did not show any disease symptoms. Inoculation with *F. graminearum* DON+ and DON− strains caused the most severe symptoms, followed by *F. graminearum* NIV+ and NIV− strains. *F. verticillioides* FUM+ and FUM− strains exhibited the lowest aggressiveness (Figure 3A). Mycotoxin-producing and nonproducing strains of the DON chemotype of *F. graminearum* and of *F. verticillioides* did not differ in aggressiveness. Inoculation with NIV-deficient mutant of *F. graminearum* led to less severe symptoms than inoculation with isogenic NIV-producing strain (*p* = 0.048, Figure 3A).

Fungal biomass was determined as the amount of fungal DNA in kernels using species-specific qPCR. *F. graminearum* (both DON+ and DON− strains) accumulated the highest biomass in planta. *F. verticillioides* (both FUM+ and FUM− strains) produced the lowest biomass, while *F. graminearum* (NIV+ or NIV−) reached intermediate levels. The biomass of wildtype strains and their mutants did not differ (Figure 3B).

The accumulation of FB1, DON, and NIV produced by *F. verticillioides* (FUM+), *F. graminearum* (DON+), and *F. graminearum* (NIV+), respectively, was quantified by HPLC-MS/MS. Mycotoxin levels (Figure 3C) followed the same trend as disease severity and fungal growth: DON was found in the highest amounts (mean 490 mg/kg), followed by NIV (mean 2.30 mg/kg) and fumonisin B1 (mean 0.15 mg/kg).

### 2.3. Disease Development after Co-Inoculation of Maize Ears with F. graminearum and F. verticillioides

For concurrent infection, maize ears were inoculated with *F. verticillioides* and *F. graminearum* 5 days after silking. Sequential inoculations were performed with the first species 5 days after silking and the second species 10 days after silking. Disease severity was evaluated 18 days after concurrent inoculation and 18 days after the second inoculation in sequential inoculation. The results are shown in Figure 4. The large number of strain combinations and inoculation methods complicates data interpretation. Therefore, we also provide a chart of relative values showing disease severity in mixed infections normalized to disease severity in plants infected with single strains (Figure 5).

In most instances, mixed inoculation with *F. verticillioides* (FUM+ or FUM−) and *F. graminearum* (DON+ or DON−), both concurrent and sequential, resulted in an increased disease severity as compared to the inoculation with *F. verticillioides* alone, and in a decreased disease severity as compared to the inoculation with *F. graminearum* alone. Concurrent and sequential inoculation with *F. graminearum* (NIV+ or NIV−) and *F. verticillioides* (FUM+ or FUM−) caused more severe disease than inoculation with *F. verticillioides* alone in 9 out of 10 strain combinations (Figure 5). Regarding the comparison with maize ears inoculated with *F. graminearum* (NIV+ or NIV−) alone, the effect of mixed inoculation did not show a consistent pattern. Concurrent and sequential inoculations comprising *F. graminearum* NIV+ did not affect disease symptoms significantly as compared to *F. graminearum* NIV+ alone. Concurrent and sequential inoculation comprising *F. graminearum* NIV- significantly increased the disease severity in three out of five cases as compared to the inoculation with *F. graminearum* NIV- only (Figure 5).

### 2.4. Fungal Biomass in Plants Co-Inoculated with Both Fusarium Species

Fungal biomass in the kernels of plants infected concurrently or sequentially with strains of both species was compared to the biomass accumulated by each strain alone (Figure 6). As in the previous section, a chart of relative values is provided to facilitate interpretation (Figure 7). Concurrent as well as sequential inoculation with *F. verticillioides* and *F. graminearum* consistently resulted in an increase in the *F. verticillioides* biomass and a decrease in the *F. graminearum* biomass as compared to inoculations with single strains (Figure 7). This effect was observed in all 20 strain combinations, and the difference was statistically significant in 10 strain combinations. The patterns of fungal biomass resembled the patterns of disease severity (Figure 5). The suppression of *F. graminearum* in mixed infection was less prominent in the strain of the NIV chemotype and its mutant. Growth suppression was observed in 8 out of 10 strain combinations, but it was statistically significant only in 4 strain combinations (Figure 7).

When *F. verticillioides* was inoculated before *F. graminearum,* suppression of *F. graminearum* was observed in all 8 strain combinations, and statistically supported in 7 of them (Figure 7). When *F. graminearum* was inoculated before *F. verticillioides*, the suppression was observed and statistically supported in 1 out of 4 strain combinations.

### 2.5. Mycotoxin Accumulation in Maize Ears Co-Inoculated with Both Fusarium Species

We compared mycotoxin levels in the kernels from maize ears inoculated with single fungal strains and co-inoculated with both species in different combinations of strains (Figure 8). As with the previous comparisons, a chart of relative differences was drawn to aid interpretation (Figure 9). Generally, mixed inoculation resulted in higher fumonisin levels and lower trichothecene levels than single-strain inoculations. The effects were consistent, except for fumonisin levels in maize ears sequentially inoculated with *F. graminearum* followed by *F. verticillioides*. The concentrations of fumonisin B1 in concurrent and sequential inoculations with *F. verticillioides* inoculated first were higher than in ears inoculated with *F. verticillioides* alone in all 8 strain combinations, and the effect was statistically significant in 4 strain combinations, in line with an increase in *F. verticillioides* biomass (Figure 7). In maize ears sequentially inoculated with *F. graminearum* followed by *F. verticillioides*, the concentration of fumonisin B1 was comparable to ears inoculated with *F. verticillioides* alone (Figure 9). The accumulation of DON and NIV was reduced in all 10 strain combinations, in line with the reduction in *F. graminearum* growth (Figure 7). Due to the large variance, the reduction was statistically significant only in 3 combinations of *F. verticillioides* with the DON-producing strain of *F. graminearum*.

## 3. Discussion

### 3.1. Saprophytic Growth of F. verticillioides Is Stimulated by F. graminearum

The suppression of *F. graminearum* growth by *F. verticillioides* in mixed cultures in saprophytic growth was expected since both fungi colonized the same space and exploited the same nutrients. Massive stimulation of the growth of *F. verticillioides* in mixed culture with *F. graminearum* (Figure 2) observed in this study was therefore surprising. We suggest that the secretion of hydrolytic enzymes by *F. graminearum*, which release nutrients from the kernels, may account for this effect. *F. graminearum* produces a rich cocktail of hydrolases: 85% of proteins identified in the secretome of *F. graminearum* growing in the presence of wheat flour were glycosidases, proteases, and esterases [63]. The main source of carbon and energy in maize kernels is starch, and amylases of *F. graminearum* are so active that they are believed to make a hidden contribution to the amylolytic potential of infected grains during malting [64]. We hypothesize that nutrients released from maize kernels by extracellular enzymes of *F. graminearum* accounted for the growth stimulation of *F. verticillioides* in mixed cultures. 

Why was stimulation of *F. verticillioides* growth by co-culturing with *F. graminearum* not observed in previous studies? Most of these studies used agar media, which could not release additional nutrients by the action of hydrolytic enzymes (cf. Figure 1). In some studies, thin layers of grains were used instead of agar, but the substrate was inoculated with a few agar disks, initiating fungal colonies that grew separately from each other until they touched. Interactions among these colonies occurred in a limited period before the termination of the cultures. In the experimental design used here, autoclaved maize kernels were inoculated with a spore suspension that provided hundreds of spores for each kernel, ensuring close interaction between the co-inoculated strains from the start of cultivation. We recommend using kernels inoculated with spores in future studies of fungal competition instead of the traditional “challenge assays” on agar-filled Petri dishes.

### 3.2. Fumonisins Appar to Contribute to the Suppression of F. graminearum by F. verticillioides in Saprophytic Cultures, whereas Trichothecenes Are Not Involved in the Interaction

Exploitation competition inadvertently takes place between *F. graminearum* and *F. verticillioides* colonizing dead plant tissue, but whether interference competition takes place is an open question. The role of DON in interference competition was suggested based on indirect evidence but not proven [35], and fumonisins inhibited the growth of *F. graminearum* in pure culture [43,44]. Our results (Figure 2) suggest that fumonisins enhance the suppression of *F. graminearum* by *F. verticillioides* in vitro. Thus, fumonisins appear to act as agents of interference competition between *F. verticillioides* and *F. graminearum.* This is presumably the case for the interactions between other fumonisin producers and their competitors, too. This result is consistent with the report that the metabolites of the fungal endophyte *Sarocladium zeae* competing with *F. verticillioides* switched off fumonisin synthesis in the latter [65]. 

The production of trichothecenes has not affected the interaction between the fungi. This is consistent with the results from the lab of Genevieve Defago, who reported that the biomass of *Trichoderma atroviride* was not affected by the ability of competing *F. graminearum* strains to produce DON [66]. The results are also consistent with the report by Müller et al. [67] that DON did not inhibit the growth of *Alternaria tenuissima* in wheat grains, and with a remarkable paper by Dawidziuk et al. [44], who reported that pure trichothecenes actually stimulated the growth of *F. verticillioides*. The growth stimulation in their work, however, was very small, and its statistical significance was marginal. In our experiments, the growth of *F. verticillioides* in the presence of *F. graminearum* strains (producing or not producing DON or NIV) was stimulated by 200% to 500%. The effects were so strong that tests of statistical significance were deemed unnecessary (Figure 2).

### 3.3. The Role of NIV as a Virulence Factor of F. graminearum in Maize Ears Was Corroborated

In the infection of maize ears, *F. graminearum* DON chemotype and its mutant were the most aggressive strains, followed by *F. graminearum* NIV chemotype and its mutant, which achieved moderate infection. Both *F. verticillioides* strains caused the mildest disease symptoms and accumulated the lowest biomass in planta. This is in line with the established ranking of *Fusarium* species by aggressiveness [6,68,69]. Lower aggressiveness of the NIV chemotype of *F. graminearum* as compared to the DON chemotype has been previously reported in winter rye [70]. 

The only mycotoxin appearing to act as a virulence factor In this study was NIV. The trichothecene-nonproducing mutant of the *F. graminearum* strain of the NIV chemotype caused fewer symptoms than the isogenic NIV producer (Figure 3A), though the biomass accumulated by both strains was similar (Figure 3B). This result corroborated the earlier finding that NIV is a virulence factor of *F. graminearum* in maize ears [25]. The observation that the NIV producer caused greater disease symptoms than its nonproducing mutant while both strains accumulated the same biomass indicates that NIV directly contributed to the disease rather than merely facilitating the colonization of plant tissue. 

Unlike in the results of Maier et al. [25], marked differences between the aggressiveness of the strains of NIV and DON chemotypes were observed in terms of disease symptoms, fungal biomass, and mycotoxin accumulation. Because Maier et al. [25] used the same strains, we assume that differences among the maize varieties used and the time allowed for disease development played a role. Gaspe Flint, used in our experiments, seems more susceptible to *F. graminearum* strain FG 2311 than the inbred line A188 used by Maier et al. [25]. Different susceptibility to DON was not responsible for this difference, as a nonproducing mutant of the strain was as aggressive as the DON producer. In an earlier study, the relative aggressiveness of *F. graminearum* (NIV producer) and *F. verticilloides* in relation to hybrid maize variety Ronaldinio and Gaspe Flint was similar: both cultivars were more susceptible to *F. graminearum* than to *F. verticilloides* [71]. Gaspe Flint was slightly more susceptible to *F. graminearum* than Ronaldinio (100% versus 85% disease severity) but less susceptible to *F. verticilloides* (20% versus 60% disease severity). The effect of mixed inoculation on pathogen growth in planta was, however, similar in all maize cultivars tested so far.

### 3.4. Trichothecenes Are Not Involved in the Stimulation of F. verticillioides Growth in Co-Infected Maize Ears

Lana Reid et al. [6] and Picot et al. [55] established that mixed infection of maize ears with *F. graminearum* and *F. verticillioides* suppressing the growth of the former while stimulating the growth of the latter. The role of trichothecenes and fumonisins in these effects has not been addressed, but the fact that trichothecenes suppress plant defenses (Section 1.2) and fumonisins inhibit fungal growth (see Section 1.3 and Figure 2), we hypothesized that these mycotoxins are involved in the interaction between *F. graminearum* and *F. verticillioides* in planta.

Our results confirmed that *F. verticillioides* suppressed disease symptoms caused by *F. graminearum* and its growth in planta, both when *F. graminearum* was inoculated earlier than or concurrently with *F. verticillioides*. A key new result is that the inoculation of maize ears with a *F. verticillioides* strain not producing fumonisins suppressed the growth of a co-infecting *F. graminearum* strain to the same degree as an isogenic fumonisin producer. If not fumonisins, what can account for the suppression of *F. graminearum* in planta? Infection with *F. verticillioides* induces the expression of many genes in maize plants, including genes involved in defense against pathogens [72]. We hypothesize that plant defenses triggered by transcriptional re-programming due to the infection with *F. verticillioides* accounted for the inhibition of *F. graminearum* in mixed infection.

### 3.5. Why Has Fumonisin-Producing F. verticillioides Strain Suppressed F. graminearum More Strongly Than Its Fumonisin-Nonproducing Mutant In Vitro but Not In Planta?

In saprophytic cultures, a *F. verticillioides* strain producing fumonisins suppressed the growth of *F. graminearum* more strongly than a fumonisin-nonproducing mutant, but this effect was not observed in co-infected maize plants. We hypothesize that the heterogeneity of infection accounts for the difference. Some kernels in inoculated cobs were likely infected with one pathogen only, and the biomass ratio for the two pathogens in mixed-infected kernels likely widely varied. The likelihood of a single kernel accumulating sufficient amounts of fumonisins to inhibit fungal growth and at the same time sufficient biomass of *F. graminearum* to detectably contribute to the total biomass of the fungus in all kernels was presumably low. This hypothesis can be tested by analyzing single kernels or even different parts of the kernels for fumonisin content and fungal biomass.

Apart from the presumed heterogeneity of mixed infection, the concentration of fumonisin B1 was lower in living tissue than in autoclaved grains (Appendix A, Table A1). This was expected because living plant tissue infected with a pathogen launches active defense responses, which is not the case with dead kernels. 

### 3.6. Why Does F. graminearum Stimulate the Growth of F. verticilliodies?

Mutual suppression of *Fusarium* spp. co-infecting grain crops has often been reported (e.g., [73,74]), but stimulation of a pathogen’s growth by co-infection with another pathogen has rarely been observed. Enhanced colonization of maize ears with *F. verticilloides* due to co-infection with *F. graminearum* is one of the few well-known cases. Our results confirmed this effect even when *F. verticillioides* was inoculated prior to *F. graminearum* (which was tested for the first time in this study), but they also showed that the stimulation of *F. verticillioides* growth was not affected by the production of DON or NIV by co-infecting *F. graminearum*. We assume that other effectors of *F. graminearum* that counteract defense responses of the plant were involved, such as the maize-specific virulence factors gramillins [75]. 

Similar plant defense responses target bacterial and fungal pathogens. It is therefore conceivable that the suppression of plant defense by *F. graminearum,* which was presumably responsible for enhanced colonization by *F. verticillioides*, also increases plant susceptibility to bacterial pathogens. Similarly, the induction of plant defense by *F. verticillioides*, which supposedly accounts for the inhibition of *F. graminearum* in plants co-infected with both pathogens (Section 3.4), might enhance plant resistance to bacterial pathogens. The interplay between fungal and bacterial pathogens co-infecting the same host plant might be a worthwhile objective for future research.

### 3.7. Genetic Background of Fungal Strains Might Explain Differences between the Suppression of DON and NIV Chemotypes 

While the stimulation of *F. verticillioides* in plants co-inoculated with the strains of the NIV chemotype was similar to the effect of strains of the DON chemotype, the suppression of *F. graminearum* in mixed inoculation was less pronounced for the NIV producer and its mutant than for the DON producer and its mutant (Figure 7). This difference was also reflected in disease severity, as no disease suppression by *F. verticillioides* was observed for the NIV-producing strain and its mutant (Figure 5). Co-inoculation with *F. verticillioides* actually enhanced disease symptoms; the effect was statistically significant in 3 out of 5 strain combinations (Figure 5). We assume that the difference in the genetic background of the two strains was the cause. This is corroborated by a previous study, which reported suppression of NIV-producing *F. graminearum* strains by *F. verticillioides* in some, but not all, combinations of strains [57].

### 3.8. What Is the Biological Function of Fumonisins in Planta?

Concurrent and sequential inoculation confirmed previously reported strong inhibition of DON-producing *F. graminearum* and its mutant in maize ears by co-infection with *F. verticillioides* in terms of fungal growth (Figure 7), disease severity (Figure 5), and mycotoxin accumulation (Figure 9). In mixed cultures in vitro, fumonisins apparently contributed to the inhibition of *F. graminearum* by *F. verticilliodes* (Section 2.1 and Section 3.2), but in maize ears, the role of fumonisins was excluded (see previous section). Fumonisins accumulate in infected kernels, especially in the endosperm and pericarp [29], with the highest amounts produced in the later stages of kernel development [76], although they are not virulence factors (Section 1.2). Fumonisins accumulate in maize ears infected with *F. verticillioides* alone, though their production is enhanced by subsequent infection with *F. graminearum* (Figure 9). These observations raise a question about the biological function of fumonisins in maize ears. 

We hypothesize that fumonisins accumulated in seeds suppress colonization with other fungi after the seeds have been shed. Protection of seeds from competitors increases the saprophytic fitness of fumonisin-producing pathogens and contributes to their overall pathogen vigor, as outlined by Audenauert et al. for DON [35]. Our hypothesis extends the concept of interference competition by a notion that protective metabolites are produced “in advance”, while kernels are still attached to the plant. This improves the protection from competitors during saprophytic growth because kernels on the ground that already contain fumonisins are better protected than kernels in which *F. verticillioides* start producing fumonisins after they have been shed. We propose that the entire life history of mycotoxin-producing fungi must be considered to decipher ecological functions of mycotoxins.

## 4. Conclusions

When *F. graminearum* and *F. verticillioides* colonize the same substrate, the growth of *F. graminearum* is suppressed, and the growth of *F. verticillioides* is stimulated. Fumonisins produced by *F. verticillioides* appear to be involved in the suppression of *F. graminearum*, acting as agents of interference competition. Co-infection of maize ears with these pathogens has the same effect on their growth, but neither fumonisins nor trichothecenes are involved. The synthesis of fumonisins in planta, which takes place with or without colonization with competitors, might improve the efficiency of the protection by loading the kernels with fumonisins before they are shed and exposed to soilborne saprophytes. 

## 5. Materials and Methods

### 5.1. Fungal Strains

Three mycotoxin-producing strains of two *Fusarium* species and their mutants deficient in the production of distinctive mycotoxins were used (Table 1): *F. graminearum* DON chemotype (DON+), its DON-deficient mutant (DON−) with disrupted *Tri5* gene, *F. graminearum* NIV chemotype (NIV+), and its NIV-deficient mutant (NIV−) with disrupted *Tri5* gene [25]. The fumonisin-producing strain of *F. verticillioides* (FUM+) and its fumonisin-deficient mutant (FUM−) with disrupted *FUM1* gene were kindly provided by Robert Proctor [77]. 

### 5.2. Fungal Media, Fungal Inoculum, Plant Material, and Growth Conditions 

Potato dextrose agar (PDA) contained 4 g potato extract, 20 g glucose, and 15 g agar per liter; pH was set to 5.7. Maize malt extract agar (MMA) contained the extract of 25 g maize grains in boiling water, 8 g malt extract, and 20 g agar per liter; pH was set to 6.4. Spore suspensions for inoculation were prepared from liquid cultures in mung bean medium as previously described [71]. Spores were counted in a Thoma chamber (0.0025 mm^2^) and adjusted to the desired concentrations using sterile tap water. Spore viability was checked on PDA. 

Mixed fungal cultures on maize grains were inoculated by adding 1 mL spore suspension containing 10^4^ viable spores of one or two strains to 20 g broken, autoclaved maize grains saturated with water in 100 mL Erlenmeyer flasks. The flasks were closed with cotton stoppers and incubated for 21 days at 24 °C in the dark without shaking. After the incubation, the cultures were frozen and freeze-dried for DNA extraction. 

Maize variety Gaspe Flint, kindly provided by Roberto Tuberosa, University of Bologna, was used in all experiments. Gaspe Flint completes its life cycle in approximately 65 days, reaches a height of 1 m, and produces ears of about 10 cm length with up to 8 rows of kernels. Before seeding, kernels were sterilized with a 4% solution of sodium hypochlorite for 15 min, followed by rinsing with sterile distilled water 3 times for 10 min. To check for a contamination, the seeds were pre-germinated in the dark on sterilized wetted filter papers at 28 °C for 5 days. Healthy seedlings were placed individually into plastic pots (13 × 13 × 11 cm), containing autoclaved mixture of topsoil and sand (*v*/*v* = 2:1), and transferred to a glass house (25 °C, 14 h photoperiod). Plants were irrigated with tap water and supplemented weekly with mineral fertilizer Hakaphos (15% N, 0.01% B, 0.02% Cu, 0.075% Fe, 0.05% Mn, 0.001% Mo, 0.015% Zn, 10% P_2_O_5_, 15% K_2_O, 2% MgO).

### 5.3. Inoculation of Plants 

Maize ears were inoculated 5 days after silking. Then, 0.5 mL sterile tap water (control plants) or spore suspension (1 × 10^5^ spores/mL) was injected into the upper part of the ear through the silk channel using a 1 mL hypodermic syringe with a blunt needle. The strains (wild types and mutants) were inoculated individually (single inoculations), concurrently (0.50 mL of each spore suspension at 1 × 10^5^ spores/mL at the same time), and sequentially (the first species 5 days after silking, the second species 10 days after silking, each as 0.50 mL of a suspension of 1 × 10^5^ spores/mL. Table 2 shows the combinations of strains used. Five ears were inoculated with each strain combination in two experiments.

### 5.4. Scoring Disease Symptoms and Sampling

Ears were harvested 18 days after inoculation, which corresponds to early dough growth stage [78], and immediately processed. Disease symptoms on dehusked maize ears, including discoloration, rotting, and/or visible fungal mycelium, were rated using a modified scale according to Reid et al. [6]. The ears were divided vertically into two symmetrical faces, and each face was further divided by a vertical and a horizontal line, resulting in four sections per face and eight sections per ear. Sections with symptoms of infection were counted, resulting in a disease index score from zero (healthy) to eight (completely infected) (Figure 10).

After scoring, the ears were detached from plants, kernels were separated from rachides, and rachides were discarded. Kernels were freeze-dried and ground in a reciprocal mill (M-400, Retsch, Haan, Germany) and stored at −20 °C.

### 5.5. DNA Extraction and Quantification of Fungal Biomass 

DNA was extracted from aliquots of 100 mg freeze-dried and ground maize kernels according to a published protocol designed to reduce the sampling error [79]. The extracts were diluted to approximately 10–15 ng µL^−1^ of total DNA by comparing the intensity of DNA bands in agarose electrophoresis with standards of known concentration and subjected to qPCR analysis as previously described [79,80]. In brief, the reaction mixture contained PCR buffer with 2.5 mM MgCl_2_, 200 µM dNTP, 0.3 µM of each primer, SYBR Green I solution (Invitrogen, Karlsruhe, Germany) diluted according to the manufacturer’s instructions, bovine serum albumin 1.0 mg/mL, and 10–15 ng total DNA. The thermocycler program for *F. verticillioides* consisted of 2 min at 95 °C, followed by 34 cycles of 40 s at 94 °C, 30 s at 62 °C, and 40 s at 72 °C, with a final extension for 4 min at 72 °C. The thermocycler program for *F. graminearum* consisted of 2 min at 95 °C, followed by 34 cycles of 30 s at 94 °C, 30 s at 64 °C, and 30 s at 72 °C, with a final extension for 5 min at 72 °C. To further minimize the error of DNA quantification, standard curves were generated using three-fold serial dilutions between 1 pg/µL and 3.3 ng/µL with two replicates, instead of common 10-fold dilutions. Each microplate contained its own standard curve. Typical standard curves are shown in Figure A2. The lowest levels of fungal DNAs in all artificially inoculated ears were at least 5 times higher than the lowest standard for qPCR. Samples with DNA levels exceeding the highest standards were diluted and re-analyzed.

### 5.6. Extraction and Quantification of Mycotoxins

Homogenized lyophilized maize flour (500 mg) was transferred to 15 mL centrifuge tubes and extracted with 5 mL acetonitrile/water (84:16, *v*/*v*) on a rotary shaker at 170 rpm overnight. After centrifugation at 4800× *g* for 10 min, 1.0 mL of the supernatant was transferred to a 2 mL reaction tube and dried at 35 °C in a vacuum concentrator. The residue was dissolved in 500 µL methanol–water (1:1, *v*/*v*); the solubilization was facilitated by sonication for 10 s and repeated with vortexing. For defetting, 800 µL of cyclohexane was added, and the samples were vortexed and centrifuged at 14,000× *g* for 10 min. Analytes recovered in the methanol–water phase were separated by HPLC (ProStar, Varian, Darmstadt, Germany) on a reverse-phase column (Kinetex, 50.0 mm × 2.1 mm, particle size 2.6 µm; Phenomenex, Aschaffenburg, Germany) at 40 °C with a methanol–water gradient, ionized by an electrospray, and detected with a mass spectrometric detector. For the analysis of DON and NIV, a triple quadrupole mass spectrometer was used as previously described [81]. Fumonisin B1 was quantified using an ion trap detector as previously described [82]. The limit of quantification (LOQ) for fumonisin B1 was 50 µg/kg; the LOQ for both NIV and DON was 100 µg/kg.

### 5.7. Statistical Analysis 

The in vitro experiment was carried out with 6 replicates. Normality of data in Figure 2 was confirmed by the Kolmogorov–Smirnov test and the significance of differences between means was determined using the two-tailed Welch *t* test, except for data obtained with the target strain F.g. NIV-, which lost 3 cultures and was therefore subjected to the Wilcoxon rank-sum test. Two in planta experiments were repeated with 5 replicates each. Relative differences between mixed and single inoculations (Figure 5, Figure 7, and Figure 9) were visualized using Sigmaplot 11.0 with DNA and mycotoxin concentrations log transformed. Statistical significance was evaluated using either ANOVA/Holm–Sidak or Kruskal–Wallis ANOVA on ranks/Dunn’s tests depending on the homogeneity of variance. The significance of differences for data presented in Figure 4, Figure 6, and Figure 8 (disease severity, fungal DNA and mycotoxin content, respectively) was computed in PAST 3.04 [83] using multiple comparisons (Mann–Whitney with Bonferroni corrections). GraphPad (Boston, MA, USA) was used for the Grubb’s test in Appendix A Table A1. For the statistical analysis, mycotoxin concentrations below the LOQ but above the limit of detection were substituted with half the LOQ value [84].

## Figures and Tables

**Figure 1 toxins-15-00575-f001:**
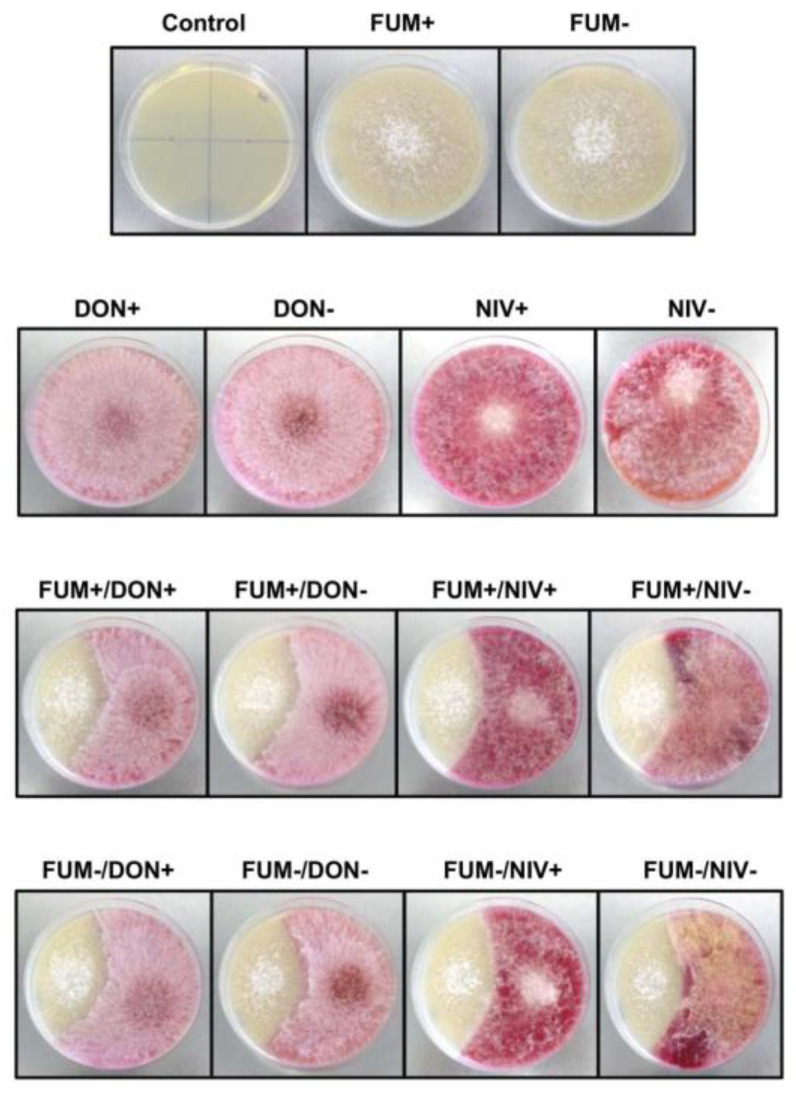
Dual cultures of *F. verticillioides* and *F. graminearum* strains producing and not producing mycotoxins. Colonies were grown on PDA for 10 days at 25 °C. FUM+, FUM−, DON+, DON−, NIV+, and NIV− designate *F. verticillioides* and *F. graminearum* strains producing and not producing the respective mycotoxins (see Section 5).

**Figure 2 toxins-15-00575-f002:**
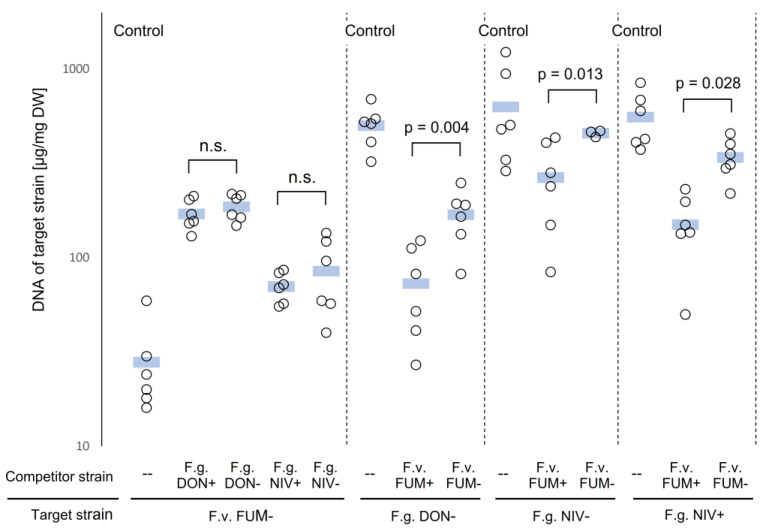
Competition between *F. verticillioides* and *F. graminearum* in vitro. Mixed cultures were grown for 21 d at 24 °C and DNA was quantified by qPCR. DW = dry weight; *F.g.* = *F. graminearum; F.v.* = *F. verticillioides*; FUM, DON, and NIV label strains producing (+) or not producing (−) respective mycotoxins. Empty circles and blue bars show data points and arithmetic means, respectively. Statistical significance was determined by two-tailed Welch test after the data passed the Kolmogorov–Smirnov test for all comparisons except the target strain *F.g.*NIV−, for which Wilcoxon rank-sum test was used. N.s. = not significant (*p* > 0.05).

**Figure 3 toxins-15-00575-f003:**
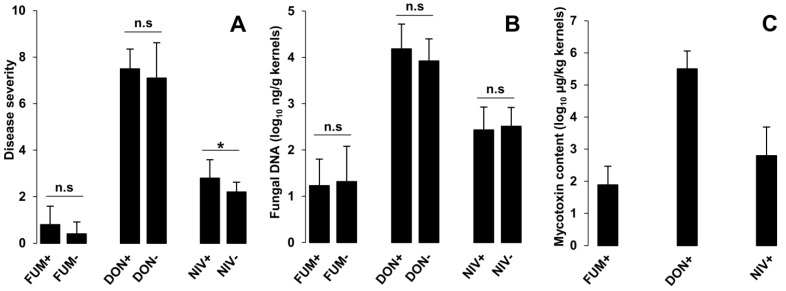
Infection of maize ears with *F. graminearum* and *F. verticillioides*. (**A**) Disease severity, (**B**) fungal DNA, (**C**) mycotoxin content. Fungal strains: FUM+ = *F. verticillioides* M-3125 (producer of fumonisins); FUM− = *Fusarium verticillioides* GfA2364 (mutant of M-3125 with disrupted fumonisin synthesis); DON+ = *F. graminearum* FG2311 (producer of DON); DON− = *F. graminearum* FG2311#2899 (mutant of FG2311 with disrupted DON synthesis); NIV+ = *F. graminearum* FG06 (producer of NIV); NIV− = *F. graminearum* FG06#7 (mutant of FG06 with disrupted NIV synthesis). Bars shows the means with standard deviations. The asterisk indicates a significant difference (*p* = 0.048) according to *t*-test; n.s. designates differences that are not significant (*p* > 0.05).

**Figure 4 toxins-15-00575-f004:**
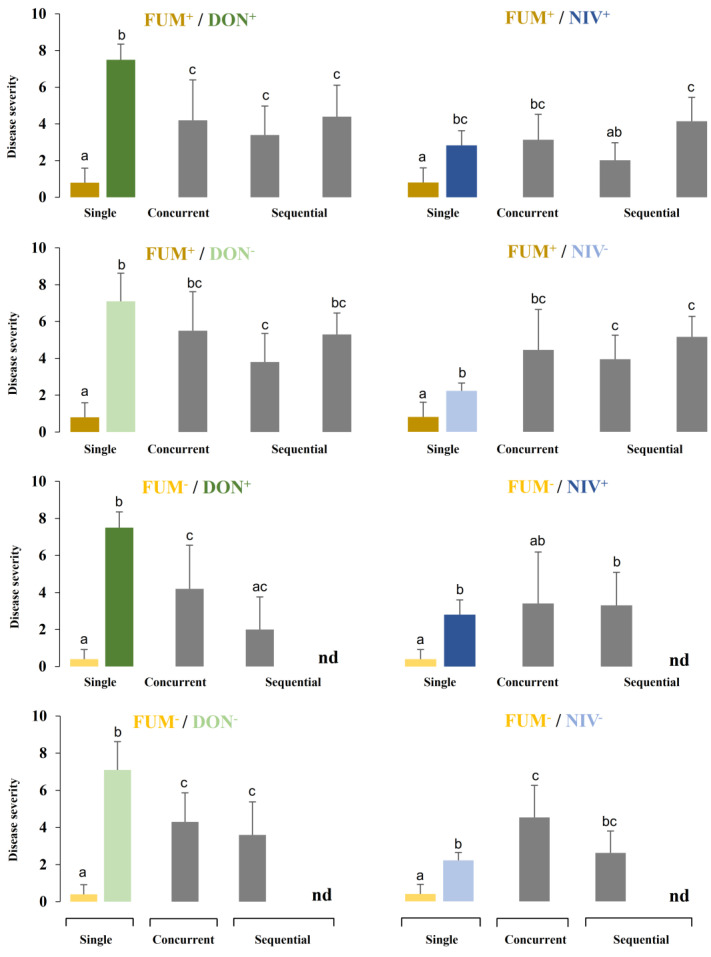
Disease severity in maize ears. Single = inoculation of a single species; concurrent = simultaneous inoculation of both species; sequential = two species were inoculated with a delay of 5 days: on the left, *F. verticillioides* followed by *F. graminearum*; on the right, vice versa. Strain labels are explained in the caption of Figure 3. Values are means; error bars show standard deviations; n.d. stands for no data. Different letters indicate significant differences (*p* < 0.05) according to Mann–Whitney pairwise test with Bonferroni correction.

**Figure 5 toxins-15-00575-f005:**
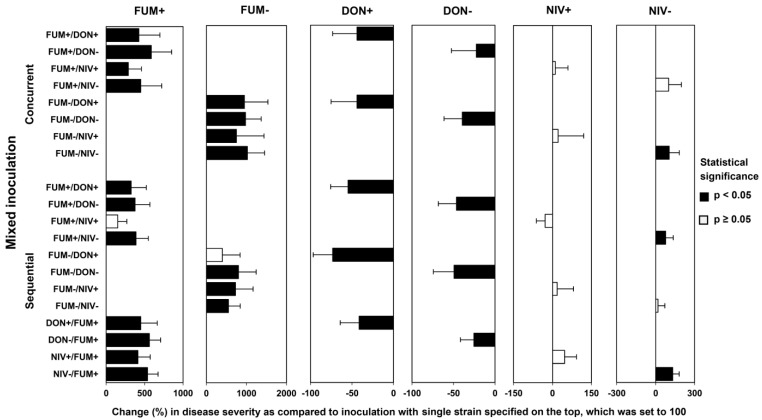
Comparison of disease severity in maize ears co-inoculated with *F. graminearum* and *F. verticillioides* (strain combination listed on the left side) with disease severity inflicted by a single strain (strains listed above the chart). Concurrent = simultaneous inoculation with both strains; sequential = the second strain was inoculated 5 days after the first strain. Strain labels are explained in the caption of Figure 3. Means of relative disease severity in % (mixed versus single-strain inoculations) with relative standard deviations are shown. Black bars indicate significant differences (*p* < 0.05) according to Kruskal–Wallis on ranks or Dunn’s test after ANOVA with Holm–Sidak correction.

**Figure 6 toxins-15-00575-f006:**
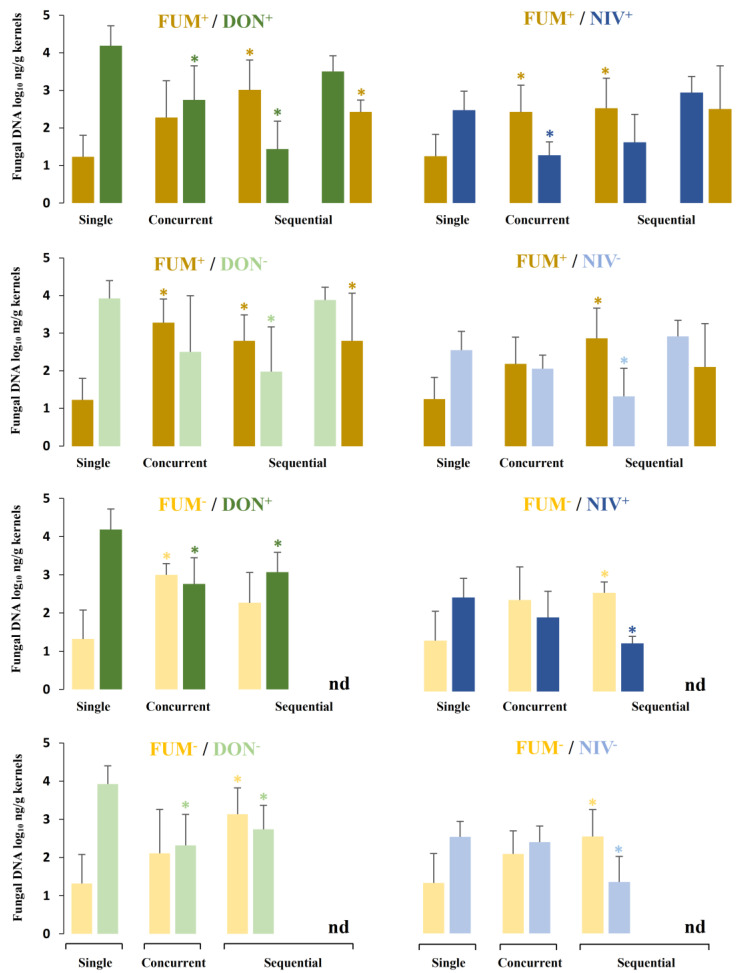
Fungal DNA in maize ears. Single = inoculation of a single species; concurrent = simultaneous inoculation of both species; sequential = two species inoculated with a delay of 5 days—on the left, *F. verticillioides* followed by *F. graminearum*; on the right, vice versa. Strain labels are explained in the caption of Figure 3. Values are the means; error bars show standard deviations; n.d. stands for no data. Asterisk indicates significant differences as compared to single-species inoculation (*p* < 0.05) according to Mann–Whitney pairwise test with Bonferroni correction.

**Figure 7 toxins-15-00575-f007:**
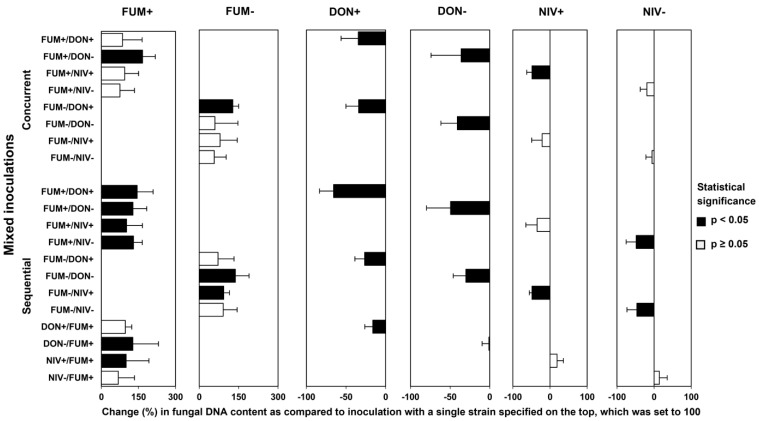
Comparison of fungal DNA content in maize ears co-inoculated with *F. graminearum* and *F. verticillioides* (strain combinations listed on the left side) to DNA accumulated after single-strain infection (strains listed above the chart). Concurrent = simultaneous inoculation with both strains; sequential = the second strain was inoculated 5 days after the first strain. Strain labels are explained in the caption of Figure 3. Means of relative values in % (mixed versus single-strain inoculations) with relative standard deviations are shown. Black bars indicate significant differences (*p* < 0.05) according to Kruskal–Wallis on ranks or Dunn’s test after ANOVA with Holm–Sidak correction.

**Figure 8 toxins-15-00575-f008:**
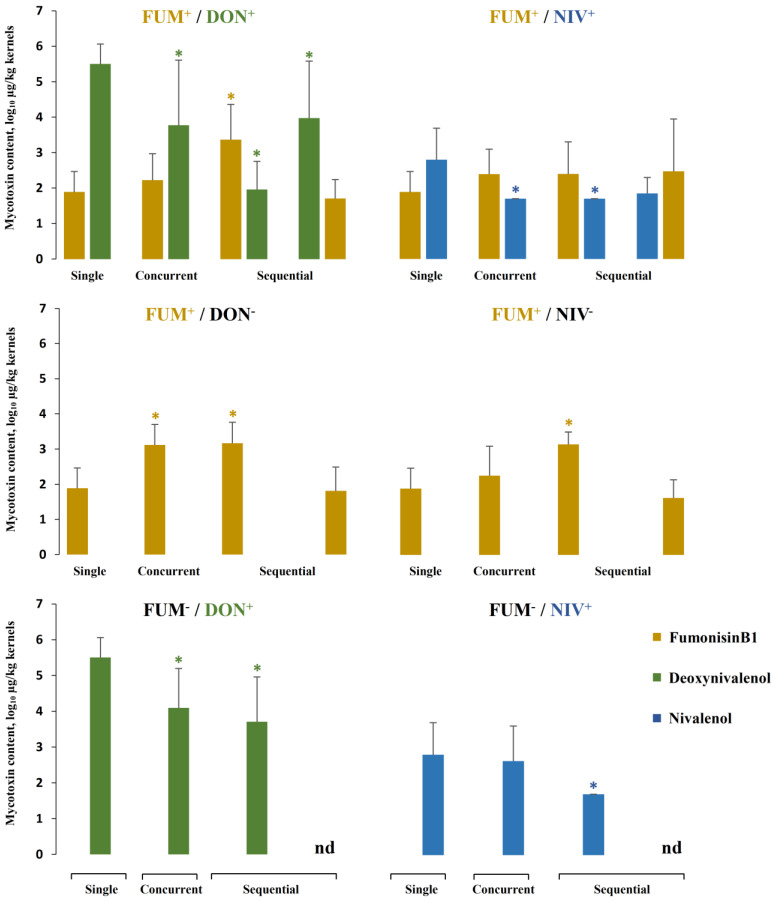
Mycotoxin content in inoculated maize ears. Single = inoculation of a single species; concurrent = simultaneous inoculation of both species; sequential = two species were inoculated with a delay of 5 days; on the left, *F. verticillioides* followed by *F. graminearum*, whereas on the right, vice versa. Strain labels are explained in the caption of Figure 3. Means with standard deviations are shown; n.d. stands for no data. Asterisks indicate significant differences as compared to the inoculation of a single strain producing the corresponding mycotoxin (*p* ≤ 0.05) according to Mann–Whitney pairwise test with Bonferroni correction.

**Figure 9 toxins-15-00575-f009:**
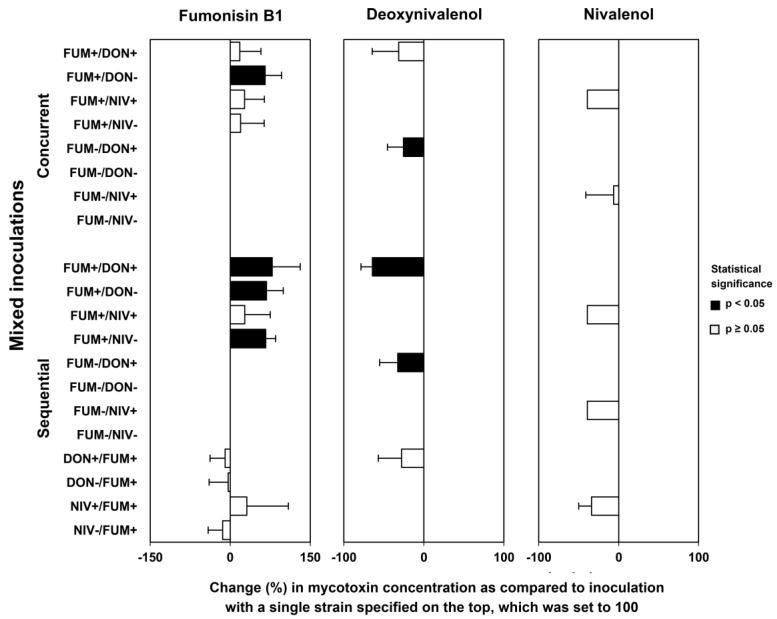
Comparison of mycotoxin content in maize ears co-inoculated with *F. graminearum* and *F. verticillioides* (strain combinations listed on the left side) to the content of mycotoxins in ears inoculated with single strains (mycotoxins listed above the chart). Concurrent = simultaneous inoculation with both strains; sequential = the second strain was inoculated 5 days after the first strain. Strain labels are explained in the caption of Figure 3. Means of relative differences with their standard deviations are shown. Black bars indicate significant differences (*p* < 0.05) according to Kruskal–Wallis on ranks or Dunn’s test after ANOVA with Holm–Sidak correction.

**Figure 10 toxins-15-00575-f010:**
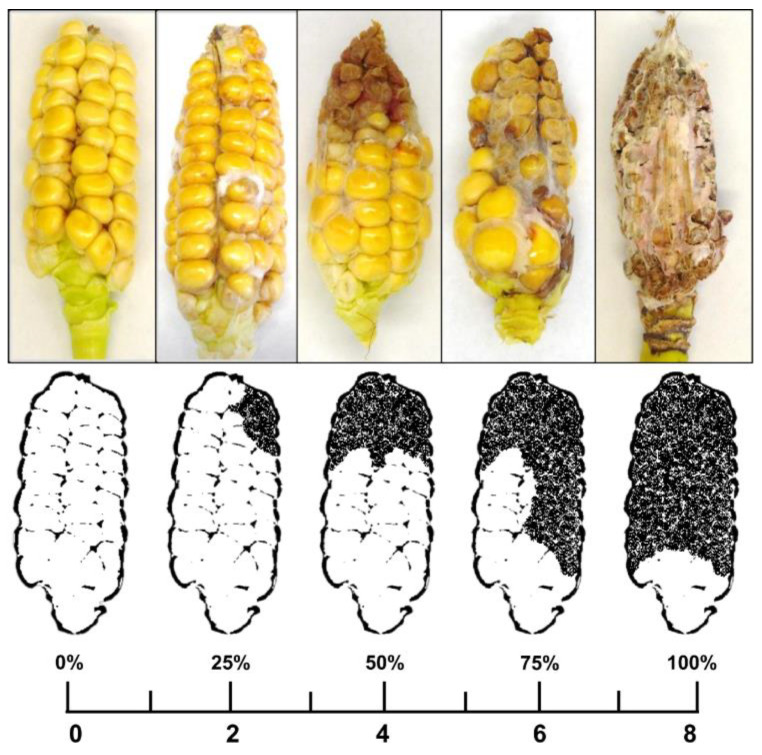
Disease severity rating of ear rot of maize. Symptoms on the front and back face of the ear were rated from zero (no symptoms) to eight (fully deteriorated).

**Table 1 toxins-15-00575-t001:** *Fungal* strains.

Strain	Code	Source
*F. graminearum* 2311	DON+	
*F. graminearum* 2311#2899 ^a^	DON−	Wilhelm Schäfer, Hamburg University, Hamburg, Germany
*F. graminearum* 06	NIV+	
*F. graminearum* 06#7 ^b^	NIV−	
*F. verticillioides* M-3125	FUM+	Robert Proctor, National Center for Agricultural Utilization Research/U. S. Department of Agriculture Peoria, IL, USA
*F. verticillioides* GfA2364 ^c^	FUM−	

^a^ Mutant of FG 2311 with disrupted *Tri5* gene [25], ^b^ Mutant of FG 06 with disrupted *Tri5* gene [25], ^c^ Mutant of M-3125 with disrupted *FUM1* gene [77].

**Table 2 toxins-15-00575-t002:** Inoculation experiments.

Inoculation Type	Fungal Strains ^a^
Single	DON+, DON−, NIV+, NIV−, FUM+, FUM-
Concurrent	FUM+/DON+, FUM+/DON−, FUM+/NIV+, FUM+/NIV−, FUM−/DON+, FUM-/DON−, FUM−/NIV+, FUM−/NIV−
Sequential ^b^	FUM+/DON+, FUM+/DON−, FUM+/NIV+, FUM+/NIV−, FUM−/DON+, FUM−/DON−, FUM−/NIV+, FUM−/NIV−, DON+/FUM+, DON−/FUM+, NIV+/FUM+, NIV−/FUM+
Control	Mock inoculation with sterile tap water

^a^ See Table 1; ^b^ The strain on the left side was inoculated first, followed by the strain on the right.

## Data Availability

Data are contained within the article and Appendix A.

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
