# Peer review of "The Role of Mycotoxins in Interactions between Fusarium graminearum and F. verticillioides Growing in Saprophytic Cultures and Co-Infecting Maize Plants"

_toxins, 2023, doi:10.3390/toxins15090575_

Round 1

Reviewer 1 Report

To elucidate the role of mycotoxins in these effects, the authors used FG mutants with disrupted synthesis of nivalenol (NIV) and deoxynivalenol (DON) and a FV mutant with disrupted synthesis of fumonisins. The results showed that 1) Fumonisins are agents of interference competition (defense metabolites) of Fusarium verticillioides, 2) Accumulation of fumonisins in planta does not suppress the growth of the co-infecting pathogen F. graminearum; but it likely enhances the protection of kernels shed on the ground from competing fungi, 3) DON or NIV are not involved in interactions between F. verticillioides and F. graminearum in planta or during saprophytic growth. The work is well organized and very important. It can be considered for publication after minor revision.

On line 42 it is: … [4-6] … , but should be … [46]. Comment: Nowadays, the middle character "–" is preferred between the numbers. Similar errors to be corrected are in many lines.

Line 7: , the growth of FV i is stimulated, the i maybe need to delete.

Line 39, 26°C, a space is required between 26 and Celsius degrees, please check the entire text.

It would be better if the chemical structural formula of the plant toxins involved in the article could be given. Such as fumonisin B1.

Can the current research findings provide assistance for effective protection against bacterial diseases. Suggest elaborating in the discussion section of the article to further highlight the importance of work.

good

Reviewer 2 Report

The manuscript titled" The role of fumonisins, deoxynivalenol, and nivalenol in interaction between Fusarium graminearum and F. verticillioides in saprophytic cultures and in maize plants". The article discussed a good point which deals with the microbe-microbe interaction. The idea is good and the work is good but I have some comments:

The Title:

The title should be changed into:

The role of mycotoxins in Fusarium graminearum and F. verticillioides interaction grown on saprophytic cultures contain maize ears and seeds.

The Abstract: the accession number of the fusarium strains and their mutants should be listed, which facilitate to other scientist to request them in further work or further cooperation.

The molecular part should be mentioned in the abstract and which genes were quantified using the Real Time PCR in compared with mycotoxin analysis.

The studied genes and their relationship with the mycotxin production or suppression. 

The suppression and induction of one of the examined mycotoxins could be quantified using HPLC or through the real time PCR.

The introduction:

1-      Too long and should be summarized.

2-      2-Should contain a small paragraph on the economic importance of the maize, the studied crop in this manuscript.

3-The citation is so crowded in some paragraphs and very low in another, please make balance.

Results

Is written good but the authors did not mentioned the role of the saprophytic media in induction or suppression of the examined fungi.

Materials and Methods:

In line 573 the primer sequence should be listed.

In line 579 Cyber green (which company and its origin) the company should be listed.

Reviewer 3 Report

This manuscript describes experiments to elucidate the role that specific mycotoxins play in the co-occurrence of two Fusarium species in maize. Specifically the co-occurrence of F. graminearum and F. verticillioides which produce deoxynivalenol or nivalenol and fumonisins respectively are explored. Two species of F. graminearum are used, one a deoxynivalenol chemotype and the other a nivalenol chemotype and observations were made of various combinations of infections using wild type and genetic mutants where the toxin biosynthetic pathways were disrupted. This study is significant in the light that co-occurrence of these two species, one normally associated with moderate climate and the other associated with warmer areas is more likely to occur in the future in maize as a result of climate change.

The manuscript is long – some 23 pages including more than 80 references - for the size of the study but the descriptions are very detailed and complete. The introduction at three pages is detailed but well organized in terms of describing what has been already published on the interaction of these two species and their mycotoxins, including some contradictory results. The experimental procedure is well designed and thorough, the results are presented in detail and plausible explanations are discussed in detail in the discussion, often in the form of questions which is unique. The conclusions are brief but adequately summarize what was presented in detail in the results and discussion. The materials and methods well describes the source of all materials and procedures used (and why).

The manuscript is very written with very few errors or typos (other than noted below) and is of excellent grammatical construction. The work adequately accounts for contradictions in data on co-occurrence previously published and provides a good prediction of what will likely be seen in the future on Fusarium contamination of maize.

Specific comments:

Abstract, line 7: Is there supposed to be an “i” after FV?

Page 3, line 144: “volatile” should be singular.

Page 12, line 345: should read “from the kernels, may account”

Page 13, line 359: “setup” is a slang term. Try “In the (experimental) design used here”

Page 15, line 487: “though” not “thought”? Or “although”

Reviewer 4 Report

In this manuscript, the authors describe the role of fumonisins, deoxynivalenol (DON), and nivalenol (NIV) in interaction between Fusarium graminearum (FG) and F. verticillioides (FV) in saprophytic cultures and in maize plants. FG stimulated the development of FV regardless of the production of DON or NIV by FG, while the growth of FG was inhibited in autoclaved kernels as well as maize ears. Fumonisin-producing FV reduced FG more effectively in autoclaved grains than a fumonisin-nonproducing strain, suggesting that fumonisins function as interference competition agents. Due to heterogeneous infection and a decreased amount of fumonisins in the planta, FG suppression in co-infected maize ears was independent of fumonisin synthesis by FV. The authors draw the conclusion that fumonisins are mediators of FV interference competition and trichothecenes have no effect on the interaction between FG and FV.

Overall, the text contains too many redundant phrases and the writing in the manuscript is really challenging to understand. I have repeatedly read to comprehend the contents.

Major points

Introduction; this section is excessively long and redundant. This section is being read to me as a review rather than as the paper's introduction.

Results; I appreciate how challenging it is to obtain relevant results in field trials such as this study. The authors need to present their data in a more efficient and organized manner so that the audience can follow along with greater ease. The presentation of most figures is challenging to comprehend in the manuscript.

Discussion; too many speculations have been made in light of the findings. For example, the authors concluded that NIV, but not DON, is a virulence factor of FG in maize ears. However, this statement cannot accepted because complemented strains were not included along with the mutants.

Minor points

L40-41. Infection of maize with F. verticillioides, the causal agent of Fusarium ear rot, results in the sporadic colonization of kernels with the appearance of whitish mycelium in the late stages.

L50-52. Rewrite this part.

L50, L56 and elsewhere. When you refer the authors’ names, just mention their family names such as Koehler and Duncan and Howard.

L54-55. Another popular inoculation method has been to inject conidia between the husks with a hypodermic needle.

L73. NIV, but not DON, acts as virulence in maize ears [25].

L84 & L85. Williams et al., Glenn et al.,

L88-91. Glenn et al., [34] provided mandatory evidence for the role of fumonisins in seedling disease by restoring fumonisin synthesis in a banana strain lacking fumonisin synthesis.

L94. ..three-week old seedlings

L240-241. The large number of strain combinations and inoculation methods complicates data interpretation.

L345. ..from the kernels

L515. Is the DON chemotype the 3ADON or the 15ADON chemotype?

L529. ..using sterile tap water

L532. ..containing 104 viable spores..

L551. ..inoculated individually (single inoculations), simultaneously…

L561. .. maize ears, including discoloration,..

L565. .. with symptoms of infections..

L566. .. completely infected..

L568. ..rachides, and rachides were discarded.

L598-599. For defatting, 800 μL of cyclohexane was added, and the samples were vortexed and centrifuged at 14,000 g for 10 min.

L621-622. For the statistical analysis, mycotoxin concentrations below the LOQ but above the limit of detection were substituted with one-half the LOQ value.

Delete Table 1. It covered in the text.

None.

Round 2

Reviewer 4 Report

I appreciate the author's responses. However, I cannot recommend the manuscript  with the minor revision to be published in the Toxins. The manuscript was written difficult to follow.

The ms is still redundant.
